# Highly Controlled Integration of Graphene Oxide into PAN Nanofiber Membranes

**Jian Hou** [1,2] , **Jaehan Yun** [2], **Sungyul Kim** [3] **and Hongsik Byun** [2,*]

1   Department of Chemical Engineering, Zibo Vocational Institute, Zibo 255314, China; houjimmy@naver.com
2   Department of Chemical Engineering, Keimyung University, Daegu 704701, Korea; ruri7220@naver.com
3   Department of Electronic and Electrical Engineering, Keimyung University, Daegu 704701, Korea; energy@gw.kmu.ac.kr
*   Correspondence: hsbyun@kmu.ac.kr; Tel.: +82-53-580-5569

**Abstract:** A highly improved strategy is established in order to systematically integrate excess exfoliated graphene oxide (GO) as fillers into polyacrylonitrile (PAN) nanofibers via electrospinning. Simple modification of GO surface allowed for their loading efficiency into the nanofibers to surpass the typical limits. Among many features, the hydrophilic and mechanical properties of these membranes were found to be significantly increased compared to the original PAN and bare GO-loaded membranes probably due to the effective reinforcing filler effect caused by the even distribution of the modified GO within the PAN nanofibers. Thus, the simple surface modification of fillers can facilitate the capability of controlling the loading efficiency into electrospun nanofibers which can highly impact the quality and performance of final composite membranes.

**Keywords:** PAN nanofibers; modified graphene oxide; electrospinning; membrane

## 1. Introduction

Nanoscale polymer fibers have been intensively investigated to implement their properties associated with high surface area, flexibility, and tunable porosity in many technical fields [1,2]. Electrospinning is one of the common approaches to prepare polymer nanofibers due to high product rate, ecofriendly manufacturing process, and low production cost [1–3]. This process involves the generation of ultrathin fibrous threads from a polymer precursor solution in electric fields, but the resulting materials prepared from bare polymers typically exhibit inherently weak chemical and physical characteristics, often limiting their practical applications. As such, these nanofibers are often modified with organic and inorganic fillers to regulate their overall properties.

Although incorporating filler materials into nanofibers can render controlled chemical physical properties, the preparation of diverse polymer composite nanofibers by electrospinning requires extensive optimization via trial and error [4,5]. Even under optimized conditions, the systematical integration of excess fillers over 10 wt % into nanofibers is still challenging [6]. Here we demonstrated the capability of loading excess graphene oxide (GO) fillers into relatively hydrophilic polyacrylonitrile (PAN) nanofibers without significant destruction of their original structures, allowing for examining the overall properties of composite membranes impacted by the excess filler loading. Given the functional groups of GO, its surface can be easily modified with a positively charged surfactant, which makes GO more compatible in a polymer precursor solution [7]. This simple modification allowed for the reliable integration of excess GO into PAN nanofibers under the same electrospinning conditions. The resulting composite membranes did not show any notable defects, possibly providing an opportunity to understand their overall properties as a function of GO content beyond typical limits.

## 2. Materials and Methods

### 2.1. Materials

Graphite flake (Bay carbon Inc. Bay city, MI., USA), sodium nitrate ($NaNO_3$, ≥99%, Sigma-Aldrich, St. Louis, MO., USA ), $KMnO_4$ (99%, Sigma-Aldrich), sulfuric acid (98%, Duksan) and hydrogen peroxide ($H_2O_2$, 35%, Samchen Co., Ltd., Seoul, Korea) were used for the synthesis of GO. Materials used to prepare nanofiber membranes were polyacrlonitrile (PAN, Mw 150,000, Sigma-Aldrich) and *N,N*-dimethyl formamide (DMF, 99.5%, Duksan, Seoul, Korea ). Cetyltrimethylammonium chloride (CTAC) as a surfactant to modify the surface of GO.

### 2.2. Preparation of Bare PAN and PAN/GO Nanofiber Membranes via Electrospinning

GO was prepared by a modified Hummer's method which is described in the previous papers [8,9]. To modify the surface of GO with surfactants, the prepared GO (0.5 wt %) was suspended in CTAC (6 wt % solution) by sonication. The mixture solution was filtered through a dead-end cell to form CTAC-modified GO sheet, followed by drying in a vacuum oven [10]. Varying amounts of GO was suspended in *N,N'*-dimethyl formamide (DMF) by sonication, followed by the addition of PAN (MW = 150 kDa) powder. The resulting homogenous solution was electrospun for 6 h under the following conditions (voltage of 15 kV, ejection speed at 0.8 mL/h, and a tip-to-collector distance of 15 cm). Such detailed information of GO including fabrication, size, thickness, and etc., were described in a previous paper [11]. As the electrospun, nanofiber sheet exhibited a low mechanical property, these fiber mats were converted to membrane sheets by a hot-press treatment [1,12].

### 2.3. Characterization of PAN-GO Composite Membranes

The structural features of membranes were examined by scanning electron microscope (SEM, JSM5410) after coating with a gold target. The thickness of membranes was estimated by a digital thickness gage. The pore diameters of membranes were analyzed with a capillary porometer (Porolux 1000) under wet and dry conditions using a Porewick standard solution. The porosity of the samples (5 cm × 5 cm) was examined by measuring the dry and wet weights of the membranes after soaking in *n*-butanol for 1 h. The wettability of membranes was examined with a contact angel analyzer (Phoenix 300) using a water droplet. The mechanical property of the membranes was evaluated by a universal tensiometer (following the ASTM D882) using a rectangular shape (500 mm/min, 100 mm x 30 mm).

## 3. Results and Discussion

Figure 1 shows the digital photos, water contact angles, and SEM images of representative bare PAN, PAN-bare GO, and a series of PAN-mGO (mGO: CATC-modified GO) composite membranes as a function of the GO content. In order to show the change color based on the various GO amount clearly picture of bare mGO sheet (Figure 1h) was prepared. It can be easily seen that the more GO in the PAN the darker color was obtained. Based on our previous strategy, the loading amount of bare GO into PAN nanofibers by electrospinning generally reached up to 4 wt % with respect to the polymer concentration due to the limited dispersity of GO in a polymer precursor solution [1]. Simply by using the surface modified GO, the GO filler was reliably loaded up to 30 wt % under the same electrospinning conditions. This surface modification involved the electrostatic interactions between the CTAC surfactant and GO which greatly improved the dispersity and compatibility of GO in polymer precursor solution as well as reliable electrospinning of nanofibers without a clogging problem. The final composite membranes loaded with excess GO did not show notable defects related to the random bead formation, which is often caused by poor dispersity of fillers [6]. The photos also show the uniform color changes from white to dark gray throughout the membranes, implying the systematical loading of GO. However, in the case of 40 wt % of GO it was found that the nanofiber mat could not be obtained due to the too much loading of GO, resulting in the difficulties of nanofiber formation [7].

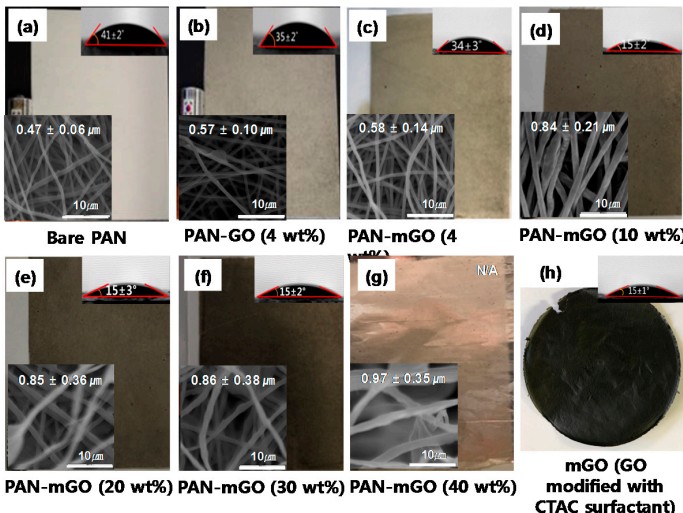

**Figure 1.** Bare polyacrylonitrile (PAN), surface-modified graphene oxide (GO), and series of PAN-GO composite membranes as a function of GO content.

So as to investigate the successful preparation of mGO the FTIR of GO, mGO, and CTAC was measured (Figure 2). It was easily recognized that mGO was successfully prepared due to the stronger specific CTAC peaks (2850 cm$^{-1}$ for C-H stretch and 2915 cm$^{-1}$ for CH$_2$) in the mGO FTIR spectra.

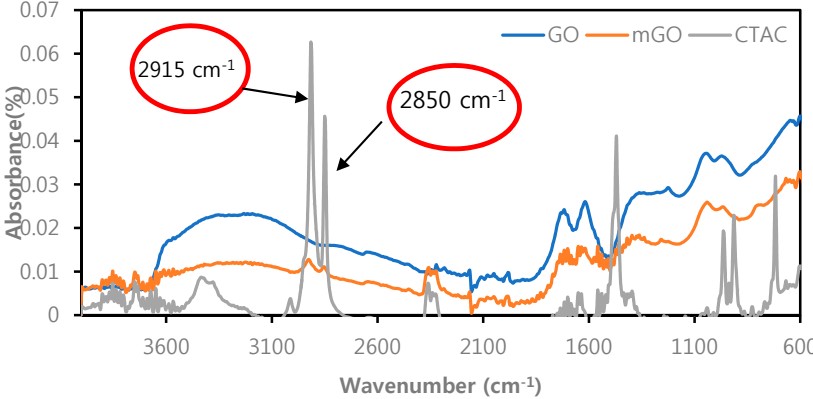

**Figure 2.** FTIR of GO, mGO, and cetyltrimethylammonium chloride (CTAC) indicates that the mGO is successfully prepared.

From the wettability test, a bare GO sheet exhibited a water contact angle of ~26° [1], and a CTAC-modified GO sheet showed an angle of ~15°. The bare PAN membrane initially exhibited a water contact angle of 41°, but the series of composite membranes systematically decreased the angle as low as 15° with the increase of GO content. The improved hydrophilicity of GO by the surface modification allowed for the excess incorporation of GO fillers into PAN nanofibers and significantly improved the wettability of the membranes. SEM images show the diameter distribution of composite nanofibers where the average diameter of nanofiber strands and its distribution gradually increased as a function of GO content. Particularly, more than 10 wt % of GO containing composite membranes led to larger diameters of the nanofibers with a wider distribution due to the systematic loading efficiency of GO into nanofibers. In addition, these composite nanofibers did not show beaded structures throughout the membrane surface.

Figure 3 shows the stress-strain curves of the bare PAN and the series of composite nanofiber membranes as a function of GO content. Unlike the composite membranes loaded with bare GO, the composite membranes integrated with the modified GO significantly improved the mechanical strength due to the reinforcement effect of GO filler [12,13]. This enhancement could be explained

by the uniform loading of GO filler throughout the PAN nanofibers where the external stress can be possibly distributed to the embedded GO to improve the overall stress [14]. Upon loading the amount of GO over 8 wt % of polymer, the composites still displayed increased strength, but notably decreased strain % possibly caused by the formation of a thicker diameter of the nanofibers and a slight reduction in the packing density [6,15].

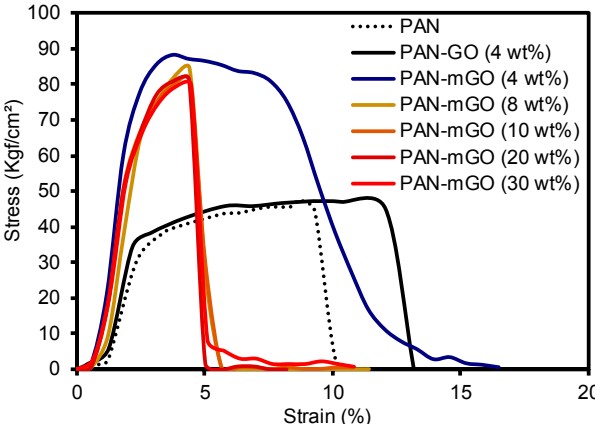

**Figure 3.** Stress-strain curves of bare PAN, and the series of PAN-GO composite membranes as a function of GO content.

To utilize the composite membranes in purification systems, the membrane integrity was evaluated as a function of GO content (Table 1). The bare PAN nanofiber membrane exhibited a pore size of 664 nm (biggest) and 167 nm (smallest). Compared to the composite membrane loaded with bare GO, all composite membranes with the modified GO exhibited notably smaller bubble points and narrower variations between the biggest pore and smallest pore size. While the bare PAN membrane possessed an average pore size of 207 nm and 80 μm thickness, the use of bare GO (4 wt %) abruptly increased the pore size to 334 nm for the composite membrane. However, the composite membranes loaded with the modified GO displayed the gradual increase of average pore size and thickness, implying the systematic incorporation of GO.

**Table 1.** Integrity of bare Bare polyacrylonitrile (PAN) and PAN-GO composite membranes as a function of graphene oxide (GO) content.

| Sample | Biggest Pore Size (nm) <Bubble Point> | Smallest Pore Size (nm) | Avg. Pore Size (nm) | Thickness (um) | Porosity (%) |
|---|---|---|---|---|---|
| Bare PAN | 664 | 167 | 207 | 80 ± 1 | 54 ± 2 |
| PAN-GO4 | 559 | 311 | 334 | 83 ± 1 | 53 ± 1 |
| PAN-mGO4 | 457 | 223 | 260 | 84 ± 2 | 51 ± 3 |
| PAN-mGO10 | 484 | 251 | 279 | 89 ± 2 | 49 ± 3 |
| PAN-mGO20 | 499 | 279 | 302 | 91 ± 3 | 48 ± 4 |
| PAN-mGO30 | 514 | 310 | 332 | 92 ± 2 | 48 ± 2 |

Interestingly, the composite membrane with the highest amount of modified GO loading still showed a pore size of 332 nm, which was slightly smaller than that of the bare GO loaded membrane. Separately, the porosity of composite membranes was marginally affected by the fillers. As such, our surface modification facilitated the successful loading of excess fillers into nanofibers without notable defects as well as changes of overall property of their resulting membranes beyond typical limits.

## 4. Conclusions

Simple modification of the GO filler significantly improved its dispersity in a polymer precursor solution during electrospinning, allowing for the systematic integration of GO fillers into PAN nanofibers without notable defects. The resulting composites offered the possibility of understanding

their structural and mechanical properties as well as membrane integrity as a function GO content well beyond typical limits. In particularly, the loading of surface-modified GO filler into polymer nanofiber membranes highly enhanced its mechanical stress and notably reduced water contact angel which are key properties applicable to water treatment systems. Thus, the proper modification of fillers can allow for a greater capability of integrating fillers into polymer nanofibers with minimal structural defects.

**Author Contributions:** J.H. and J.Y. designed the experiments. J.H. and J.Y. performed the experiments. J.H. and S.K. analyzed the data. J.H. and H.B. wrote the manuscript.

**Funding:** This research is supported by the National Research Foundation of Korea Grant funded by the Korean Government (NRF-2018R1A2B6008854).

**Conflicts of Interest:** The authors declare no conflict of interest.

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
