# Peer review of "Highly Controlled Integration of Graphene Oxide into PAN Nanofiber Membranes"

_applsci, doi:10.3390/app9050962_

Round 1
Reviewer 1 Report
The paper by J. Hou et al. is interesting but it should be greatly improved before publication
in Applied sciences.
1- In Fig. I PAN mGO (40%) and mGO modified by GTAC are not discuussed in text of the paper ; more information should be given about the construction of these samples notably (h) view. Why the colour is more clear than (f)
2- The fabrication of the Graphene Oxide nanoparticles should be more precise. The characterisation of these objects should be given: size and zeta potential. This value
should be given also after grafting CTAC.
3- A quantitative determination of the amount of CTAC fixed onto nano carbon should be given. Is this compound fixed with high stability on the carbon nanoparticles?
This qestion gives also the rise of another question: the films of PAN mGO should be studied for their stability in water: some CTAC could be released and can deliver CTAC into water as
there is no covalent link with the carbon.
4- In table I
Last line : PAN mGO 30 should be written
The colum thickness should contain a domain +- value instead of approximately
5- As the precise procedure of building up the film is not known it should be discussed on the increase of thickness. Is the weight of the sample constant or increasing?
Author Response
Applied Chemistry
Manuscript ID: applsci-458064
Type of manuscript: Article
Title: Highly Controlled Integration of Graphene Oxide into PAN Nanofiber
Membranes
Dear Reviewer#1
Thank you very much for handling the above-referenced manuscript and for your critical feedback to consider it further. Attached, please find a revised version with a detailed list of changes below.
In this revision, we have tried to amend our manuscript to clarify all the concerns raised by the reviewers. We hope that you will consider publishing our revised manuscript without the need for further review. Thank you again for your time and consideration.
Sincerely,
Hongsik Byun
Professor, Dept. Chemical Engineering
Keimyung University, Deagu, Korea
President, The Aseanian Membrane Society
Senior Advisor, The Korean Membrane Society
Chairman, Membrare Co. Ltd., Korea
www.membrare.com
Tel: +8253580-5569
Fax: +8253590-6305

Reviewer 2 Report
In the paper "Highly Controlled Integration of Graphene Oxide into PAN Nanofiber Membranes" authors perform surface modification with a surfactant on graphene oxide in order to facilitate its incorporation and dispersion in PAN solution. After this, nanofibres, were manufactured via electrospinning.
Overall the paper is well presented, easy to follow and conclusions are supported by the results. Only minor issues are a couple of lines were a word is repeated two times, like in line 113.
After those minor issues are corrected, the work can be published.
Author Response
Applied Chemistry
Manuscript ID: applsci-458064
Type of manuscript: Article
Title: Highly Controlled Integration of Graphene Oxide into PAN Nanofiber
Membranes
Dear Reviewer #2:
Thank you very much for handling the above-referenced manuscript and for your critical feedback to consider it further. Attached, please find a revised version with a detailed list of changes below.
In this revision, we have tried to amend our manuscript to clarify all the concerns raised by the reviewers. We hope that you will consider publishing our revised manuscript without the need for further review. Thank you again for your time and consideration.
Sincerely,
Hongsik Byun
Professor, Dept. Chemical Engineering
Keimyung University, Deagu, Korea
President, The Aseanian Membrane Society
Senior Advisor, The Korean Membrane Society
Chairman, Membrare Co. Ltd., Korea
www.membrare.com
Tel: +8253580-5569
Fax: +8253590-6305

Reviewer 3 Report
The article reports the effect of modification of GO with a surfactant on the hydrophilic and mechanical properties of PAN/GO nanofiber membranes prepared via an electrospinning method. The results are presented accurately and well discussed. However, more details must be given to make them informative for researchers in the same fields according to the following questions raised during the manuscript reading.
Page 1. Row 48: “KMnO4” is in a different font than surrounding text.
Page 1. Row 41. Relevant bibliographical references about surfactant modified graphene oxide must be included.
Page 2, row 52: the word “cetyltrimethylammonium” should be capitalized.
Page 2, row 55: the amount of CTAC used to modified GO should be mentioned.
The words “cetyltrimethylammonium chloride” must be omitted as the acronym (CTAC) has already been defined in line 52.
Page 2, row 71: the authors should give information about the cross head speed chosen to determine the tensile properties.
“(100 mm x 30 mm)” is in a different font than surrounding text.
Page 4, row 113: the word “utilize” is repeated twice, one after the other.
The structural and morphological characterization of graphenic materials is missed. The authors should investigate it by SEM and XRD techniques. PAN/GO and PAN/modified-GO composites characterization by XRD it would also be necessary.
Author Response
Applied Chemistry
Manuscript ID: applsci-458064
Type of manuscript: Article
Title: Highly Controlled Integration of Graphene Oxide into PAN Nanofiber
Membranes
Dear Reviwer #3:
Thank you very much for handling the above-referenced manuscript and for your critical feedback to consider it further. Attached, please find a revised version with a detailed list of changes below.
In this revision, we have tried to amend our manuscript to clarify all the concerns raised by the reviewers. We hope that you will consider publishing our revised manuscript without the need for further review. Thank you again for your time and consideration.
Sincerely,
Hongsik Byun
Professor, Dept. Chemical Engineering
Keimyung University, Deagu, Korea
President, The Aseanian Membrane Society
Senior Advisor, The Korean Membrane Society
Chairman, Membrare Co. Ltd., Korea
www.membrare.com
Tel: +8253580-5569
Fax: +8253590-6305

Round 2
Reviewer 1 Report
The paper by J. Hou is now acceptable for publication in Applied Sciences.
Reviewer 3 Report
The authors have satisfactorily responded to all my questions and made the necessary changes to the manuscript.